# Colorectal Cancer Heterogeneity and the Impact on Precision Medicine and Therapy Efficacy

**DOI:** 10.3390/biomedicines10051035

**Published:** 2022-04-30

**Authors:** Gerardo Rosati, Giuseppe Aprile, Alfredo Colombo, Stefano Cordio, Marianna Giampaglia, Alessandro Cappetta, Concetta Maria Porretto, Alfonso De Stefano, Domenico Bilancia, Antonio Avallone

**Affiliations:** 1Medical Oncology Unit, “S. Carlo” Hospital, 85100 Potenza, Italy; marianna.giampaglia@ospedalesancarlo.it (M.G.); domenico.bilancia@ospedalesancarlo.it (D.B.); 2Department of Oncology, “San Bortolo” General Hospital, Azienda ULSS8 Berica, 36100 Vicenza, Italy; giuseppe.aprile@aulss8.veneto.it (G.A.); alessandro.cappetta@aulss8.veneto.it (A.C.); 3Medical Oncology Unit, CDC Macchiarella, 90138 Palermo, Italy; oncologia@casadicuramacchiarella.it (A.C.); oncologia2@casadicuramacchiarella.it (C.M.P.); 4Medical Oncology Unit, “Maria Paternò Arezzo” Hospital, 97100 Ragusa, Italy; stefano.cordio@asp.rg.it; 5Experimental Clinical Abdominal Oncology Unit, Istituto Nazionale Tumori IRCCS-Fondazione “G. Pascale”, 80121 Napoli, Italy; a.destefano@istitutotumori.na.it

**Keywords:** metastatic colorectal cancer, precision medicine, *RAS* and *BRAF* inhibitors, immunotherapy, anti-*HER-2*, *TRK* inhibitors

## Abstract

Novel targeted therapies for metastatic colorectal cancer are needed to personalize treatments by guiding specific biomarkers selected on the genetic profile of patients. *RAS* and *BRAF* inhibitors have been developed for patients who become unresponsive to standard therapies. Sotorasib and adagrasib showed promising results in phase I/II basket trial and a phase III trial was planned with a combination of these *RAS* inhibitors and anti-*EGFR* monoclonal antibodies. Encorafenib and binimetinib were administered in phase II clinical trials for *BRAF* mutated patients. Pembrolizumab is now recommended in patients exhibiting microsatellite instability. Larotrectinib and entrectinib showed a fast and durable response with few and reversible adverse events in cases with *NTRK* fusions. Trastuzumab and trastuzumab deruxtecan exhibited promising and durable activity in *HER-2*-positive patients. In this review, the reasons for an extension of the molecular profile of patients were assessed and placed in the context of the advancements in the understanding of genetics. We highlight the differential effect of new targeted therapies through an ever-deeper characterization of tumor tissue. An overview of ongoing clinical trials is also provided.

## 1. Introduction

Colorectal cancer (CRC) is a major global health issue, being the third most commonly diagnosed malignancy with an estimated number of more than 1.9 million new cases and about 935,000 deaths worldwide in 2020 [1,2]. In twenty percent of cases, the disease occurs at an advanced stage at diagnosis, while up to 50% of patients with early-stage disease relapse, despite curative surgery, adjuvant chemotherapy, and/or radiation therapy. Advances in multidisciplinary treatment and care have led to significant improvements in survival, but a cure is not possible for most of these patients [3].

Targeted therapies work on cancer cells by directly inhibiting cell proliferation, differentiation, and migration. The tumor microenvironment, including local blood vessels and immune cells, could also be altered by targeted drugs, so as to inhibit tumor growth. Various pathways that mediate CRC initiation, progression, and migration, as well as those that can activate signaling cascades, are ideal sites for these drugs [4].

In recent years, clinical outcomes in metastatic colorectal patients (mCRC) have thus improved significantly. International guidelines have now first of all mandated, as a standard of care, the identification of approximately 40% of patients without rat sarcoma virus (*RAS*) and B-rapidly accelerated fibrosarcoma (*BRAF*) oncogene mutations [5]. The anti-epidermal growth factor receptor (*EGFR*) monoclonal antibodies cetuximab and panitumumab, combined with chemotherapy, are standard treatments in these cases. The most important limitation of these drugs is in inducing resistance sooner or later [6,7], although evidence suggests that some patients may benefit from a rechallenge strategy in the course of their disease [8]. On the contrary, patients with mutated tumors benefit from anti-vascular endothelial growth factor (*VEGF*) agents (bevacizumab and aflibercept) when combined with chemotherapy [9,10].

The refinement of the knowledge of molecular biology and new agents capable of more specifically targeting previously unknown mutated genes change the therapeutic perspectives of many patients. Since 2013, the National Cancer Institute (NCI) has launched, with renewed interest, programs, and initiatives to deepen the function of *RAS* and learn about its biology to identify innovative drugs [11]. Although *BRAF* inhibitors as single agents have shown only modest activity in *BRAF*-mutated mCRC, several clinical trials have demonstrated that combination therapies with *EGFR* and mitogen-activated protein kinase (*MEK*) inhibitors overcome resistance mechanisms [12]. Tumors with deficiency in mismatch repair genes (*dMMR*) are highly responsive to immune checkpoint inhibitors both in first- and second-line settings [13]. Human Epidermal Growth Factor Receptor (*HER-2*) amplification has emerged as a promising therapeutic target for mCRC patients displaying this molecular abnormality. Patients with *RAS* wild-type (wt) and *HER-2* overexpression have been successfully treated with anti-*HER-2* antibodies [14]. Other emerging actionable molecular alterations include rare gene fusions of neurotrophic tyrosine kinase receptor, type 1 (*NTRK1*) that can be targeted by specific inhibitors [15].

Although new techniques such as next-generation sequencing (*NGS*) and the availability of tumor panels allow the identification of many predictive markers (Figure 1), their application in clinical practice is often difficult. It is quite evident that recommendations are needed to guide the physician in these cases and support therapeutic decision making for patients with mCRC. 

This paper provides an overview of existing CRC-targeted agents (Table 1) and their underlying mechanisms, as well as a discussion of their limitations and future trends.

## 2. Methods

We carried out a fine literature research to find relevant publications from the last few years utilizing the PubMed database (www.ncbi.nlm.nih.gov/pubmed/, accessed on 31 January 2022) by entering the following terms: “metastatic colorectal cancer” and “biomarkers”. Full-text manuscripts reporting data on these items were searched and reviewed in detail. In addition, we performed a general excursus of the principal oncology congresses to identify abstracts reporting new mCRC biomarker data and published between 1 January 2016 and December 2021.

## 3. Mutational Status of *RAS*

*RAS* is a family of proteins expressed on all cells and responsible for transmitting signals through which proliferation, adhesion, migration, and cell differentiation as well as the apoptosis process are stimulated and controlled. When these proteins mutate, the cells acquire properties of invasion and metastatization. The main mutations concern Kirsten rat sarcoma virus (*KRAS*) and neuroblastoma ras viral oncogene homolog (*NRAS*) and consist in the change of a single nucleotide or its deletion or insertion into a deoxyribonucleic acid (*DNA*) sequence [23].

*KRAS* mutations are found in approximately 40% of patients with *mCRC*, mainly in exon 2, codons 12 (70–80%) and 13 (15–20%), less frequently in exons 3 and 4. The most common *KRAS* mutations of exon 2 (codons 12 and 13) include *G12D* (32.4%), *G13D* (14.1%), *G12V* (11.3%), *G12S* (9.9%), *G12C* (8.5%), and *G12A* (2.8%) [24]. Regardless of type and location, *KRAS* mutations play a prognostic role and, when grouped together, patients with *KRAS* mutated metastatic disease have a higher mortality rate (18.5% vs. 34%) and a shorter survival (14 months vs. 23.5 months) than wt ones [25]. Moreover, the various mutations of *KRAS* are not equal to each other and a pooled analysis of five randomized trials showed that they were associated with heterogeneous outcomes [26]. While patients harboring the *KRAS G12C*-variant correlated with inferior overall survival (OS) compared with unmutated tumors and a similar trend for OS was seen in the *KRAS G13D*-variant, more frequent *KRAS* exon 2 variants like *G12D* and *G12V* did not have a significant impact on OS. Although the reasons are not clear, these biological differences could be explained by a separate activation process for each individual variant of the *KRAS*-depending pathways.

*NRAS* mutations in codons 2, 3, and 4 are rare and found in 3–5% of metastatic patients, more frequently in left-side colon and mainly in women. An Italian study has shown that *NRAS* and *KRAS* mutated tumors did not show significant differences in terms of clinical and pathological characteristics, except for a lower prevalence of mucinous histology and lung metastases among *NRAS* mutated tumors. In uni- and multivariate analysis, *NRAS* mutations were associated with shorter OS than in all wt patients (median OS 25.6 vs. 42.7 months) [27]. *NRAS* mutations recorded at exon 3 identify patients with markedly lower OS not only compared to wt ones (HR 2.85; *p* < 0.01), but also to those with mutations in exon 2 (HR 2.0; *p* = 0.039) [28].

*EGFR* is a membrane receptor tyrosine kinase and is a key target for monoclonal antibodies which bind on the extracellular domain of the receptor. Several phase II and III trials indicate that an increased gene copy number of *EGFR* or mutations of *KRAS* and *NRAS*, responsible for downstream signalling, are important determinants of response to cetuximab and panitumumab [29,30,31,32]. While an improvement of treatment efficacy is proven only in wt patients, *RAS* mutated patients either had no benefit from the addition of anti-*EGFRs* or even showed a worse outcome than their comparators [7,33]. Thus, since 2013 extended *RAS* analysis is recommended at the time of diagnosis in metastatic disease for all patients [5,34].

The best treatment for *RAS* mutated patients is not sufficiently standardized due to the lack of clinical trials specifically designed for these patients. Although the major international guidelines report that chemotherapy plus bevacizumab should be the preferred first-line therapy for patients with *RAS* mutation, the evidence is debatable [35,36]. First of all, there are no prospective randomized trials for this specific setting of patients. Secondly, the benefit of the addition of bevacizumab to first-line treatment significantly prolonged progression-free survival (PFS), while in *RAS* mutated patients it led to a relatively modest reduction in risk of death of 12%. Thirdly, data from randomized phase III studies are confusing as they include patients with *RAS* wt and *RAS* mutated disease. Only recently, a Chinese randomized study specifically enrolled patients with *RAS* mutation and with metastases limited exclusively to the liver showing that bevacizumab plus chemotherapy versus chemotherapy alone results in a higher conversion rate in liver surgery, an increase in response rate (RR), and an extension of PFS and OS [37]. The strategy of employing triplet chemotherapy [leucovorin, oxaliplatin, irinotecan, and fluorouracil (FOLFOXIRI)] plus bevacizumab also appears to confirm that mutated *RAS* patients have no improvement in OS, although this is partially confirmed by the TRIBE2 study including many patients with these characteristics [38,39].

Only in recent years the clinical research focused on the possibility of its direct inhibition with specific target therapies. Few molecules have reached clinical development more directly and many have instead been studied in the preclinical phase [40].

Focusing only on the former, sotorasib (AMG 510) and adagrasib appear to be very promising. The *KRAS* p.G12C mutation occurs in approximately 1 to 3% of CRC and increases the proliferation and survival of cancer cells. The small molecules sotorasib and adagrasib specifically and irreversibly inhibit *KRAS^G12C^*. A phase I study has shown that in 42 patients with heavily pretreated mCRC, sotorasib, administered orally once a day at a dose of 960 mg, was tolerable as only 15 patients (11.6%) reported grade 3 or 4 treatment-related adverse events (AEs) [16]. The most common events were diarrhea (29.5%), fatigue (23.3%), and nausea (20.9%). The median duration of treatment was 3.9 months and, although a confirmed partial response (PR) was observed in only 7.1% of patients, 28 of them (66.7%) had stable disease (SD). The median PFS was 4.0 months. In the same way, adagrasib was tested in a study involving 17 patients, mostly non-small cell lung cancer or mCRC. The reported toxicity was low once again and 12 patients were evaluated for the response: four had a PR and eight had SD. All responders received a dose of 600 mg twice a day; one patient with *mCRC* had a 47% reduction, six weeks after the start of treatment [41]. 

However, irreversible inhibition of *KRAS^G12C^* may result in another benefit in patients due to the signaling pathways that determine a susceptibility to *KRAS* feedback reactivation. *EGFR* signaling is involved in this reactivation, providing a rational co-targeting strategy for *KRAS*-mutant *mCRC*.

KRYSTAL-1 (NCT03785249), presented at ESMO 2021, is a multicohort study evaluating adagrasib in patients with *KRAS^G12C^*-mutant advanced solid tumors [17]. The cohort of *mCRC* patients was treated with adagrasib 600 mg twice a day as monotherapy or in association with cetuximab. Forty-six patients received adagrasib monotherapy and, among the 45 evaluable for clinical activity, the RR was 22% and the disease control rate (DCR) was 87%. Median duration of response was 4.2 months and median PFS was 5.6 months. Thirty-two patients were treated with adagrasib + cetuximab. Among the 28 patients evaluable for clinical activity, the RR was 43% and DCR was 100%. AEs of any grade occurred in 91% and grade 3–4 events in 30% of patients treated with adagrasib monotherapy, while AEs of any grade occurred in 100% and grade 3–4 events in 16% of patients with combination therapy. Adagrasib + cetuximab is currently being evaluated as second-line therapy in a phase III study in this setting of patients (NCT04793958). Sotorasib and panitumumab will also be evaluated in another phase III study (EudraCT Number: 2021-004008-16) in which the combination of the two drugs will be compared versus regorafenib or trifluridine-tipiracil (TAS-102) in pretreated patients with the same mutation.

*RAS* mutations in *mCRC* are associated with aberrations in *DNA* replication and this would make tumors sensitive to inhibition of *WEE1*, which is a tyrosine protein kinase that regulates G_2_/M checkpoints in the cell cycle in response to *DNA* damage. A randomized phase II trial (Focus4-C) has tested the hypothesis that adavosertib (AZD1775), a potent oral *WEE1* inhibitor, could increase PFS in patients who had either a clinical response or a SD after at least 16 weeks of chemotherapy administration compared to those undergoing active monitoring [42]. Adavosertib was associated with a PFS improvement over active monitoring (median 3.61 vs. 1.87 months; HR = 0.35; 95% CI, 0.18 to 0.68; *p* = 0.0022), while the OS did not show substantial differences (median 14.0 vs. 12.8 months; HR = 0.92; 95% CI, 0.44 to 1.94; *p* = 0.93). In prespecified subgroup analysis, adavosertib activity was greater in left-sided tumors (HR = 0.24; 95% CI, 0.11 to 0.51), versus right-sided (HR = 1.02; 95% CI, 0.41 to 2.56; interaction *p* = 0.043). Adavosertib was well-tolerated, and few patients reported grade 3 toxicities such as diarrhea (9%), nausea (5%), and neutropenia (7%).

Immunotherapy may be an option for *RAS* mutated patients. *RAS* mutation influences the tumor microenvironment, the reduction in the number of tumor-infiltrating lymphocytes (*TILs*), mismatch-repair defects, and the increase in the expression of programmed cell death ligand 1 (*PD-L1*) [11]. A preclinical study has demonstrated the potential of therapy with *TILs* capable of recognizing tumor cells harboring the *KRAS^G12D^* mutation resulting in tumor regression and a PR lasting 9 months in a patient with *mCRC* [43].

Another way to achieve *RAS* inhibition may be to antagonize other proteins of its pathway, as in the case of mitogen-activated protein kinase (*MAPK*). It has been shown in preclinical models that the combination of *MEK* inhibitors with an anti-*PD-L1* results in a synergistic and lasting tumor regression [44]. Consequently, the feasibility of a combination of atezolizumab and the *MEK* inhibitor cobimetinib has been evaluated in a phase Ib study [45]. The primary aims were safety and tolerability. In most patients, cobimetinib was administered once daily orally for 21 days on, 7 days off, while atezolizumab was dosed at 800 mg intravenously every 2 weeks. The most common AEs related to the treatment were diarrhea (67%), rash (48%), and fatigue (40%), without significant differences with single-agent cobimeitinib and atezolizumab. Confirmed responses were obtained in 7 out of 84 patients (8%), a SD in 19 cases (23%). The 12-month PFS and OS rates were 11% and 43%. Nevertheless, this potential synergistic activity was not confirmed in a subsequent phase III study designed to compare atezolizumab plus cobimetinib or atezolizumab monotherapy versus standard therapy in the third-line setting [46]. In fact, IMblaze370 did not meet its primary endpoint of improved OS with experimental arms versus regorafenib.

The association of cytotoxics and bevacizumab may promote the sensitivity to immunotherapy increasing the exposure of neoantigens, inducing immunogenic cell death, and increasing the immune infiltration in tumor microenvironment [47]. AtezoTRIBE, a prospective, open label, phase II, comparative trial, randomized 218 initially unresectable mCRC patients (over 70% *RAS* mutated), irrespective of mismatch repair (*MMR*) status, to receive up to 8 cycles of FOLFOXIRI/bevacizumab (arm A) or FOLFOXIRI/bevacizumab/atezolizumab (arm B), followed by maintenance with 5-FU/bevacizumab or 5FU/bevacizumab/atezolizumab until disease progression [48]. The primary endpoint was PFS. A significant advantage with the addition of atezolizumab was observed in PFS (13.1 vs. 11.5 mos, HR 0.69, *p* = 0.012), but not in ORR (59% vs. 64%, *p* = 0.412). No safety issues were reported. 

## 4. Mutational Status of *BRAF*

The routine molecular characterization of mCRC patients includes, beyond *RAS*, the tumor *BRAF* mutational testing, according to the recommendations provided by the international clinical guidelines [49,50,51]. The *BRAF* gene encodes a serine-threonine protein kinase that is part of the *MAPK* pathway. *BRAF* mutations occur in about 10% of patients with *mCRC* and are usually mutually exclusive with *RAS* mutations. They are most frequently caused (>90%) by the replacement of valine with glutamic acid inside the 600 codon (*BRAF^V600E^*), leading to an overactive *MAPK* pathway [12,52]. 

The presence of somatic *BRAF^V600E^* alteration mostly characterizes a subgroup of mCRC patients associated with the female sex, right-sided colonic cancer, mucinous histology, microsatellite instability (*MSI*)/*dMMR* profile and metastatic spread mainly to lymph nodes and peritoneum [12,52,53]. *BRAF^V600E^* mutant mCRC patients show a shorter OS and achieve a very modest benefit from standard chemotherapy, highlighting their poor prognosis [12]. Moreover, the benefit of anti-*EGFRs* in these cases remains unclear and two meta-analyses have not been able to provide more clarity to the issue [54,55]. With the advent of next-generation sequencing, non-*BRAF^V600E^* mutations have been increasingly identified in clinical practice, more often observed in younger patients, males and showing fewer peritoneal metastases compared to *BRAF^V600E^* mutants [56]. The expected survival of this subgroup of patients is not negatively influenced, as happens for *BRAF^V600^*. Of note, most of the non-*BRAF^V600E^* mutations, in particular those belonging to class 3, retain a sensitivity to anti-*EGFRs* based treatments [57]. 

The optimal treatment of *BRAF^V600E^* mutant mCRC patients has been matter of active clinical research and controversial debate. In recent years, FOLFOXIRI plus bevacizumab was introduced as a standard of care for initial treatment of this subgroup of mCRC patients [49,50,58]. The use of this intensive combination was mainly supported by a subgroup analysis of 28 *BRAF^V600E^* mCRC patients enrolled in the TRIBE trial, which showed a median OS of 19.0 months in patients treated with FOLFOXIRI/bevacizumab, whereas patients treated with leucovorin, irinotecan and fluorouracil (FOLFIRI)/bevacizumab had a shorter median OS of 10.7 months [59]. However, the evidence of benefit from the intensified approach was not confirmed in the TRIBE 2 trial, in which patients were randomized to FOLFOXIRI plus bevacizumab or to leucovorin, oxaliplatin and fluorouracil (FOLFOX) plus bevacizumab [38]. In addition, a recent meta-analysis of five randomized trials comparing FOLFOXIRI plus bevacizumab to a doublet combination plus bevacizumab confirmed the absence of any advantage of FOLFOXIRI plus bevacizumab in *BRAF* mutated cases [60]. Based on this evidence, an intensification of treatment does not offer a clear benefit in the frontline treatment of *BRAF^V600E^* mutated mCRC patients. However, patients with *BRAF^V600E^* mutation tumors appear to benefit from anti-*VEGF* therapy, unlike that with anti-*EGFRs*, similarly to patients with *BRAF* wt tumors [12].

A major efficacy of an antiangiogenic agent in combination with chemotherapy has also been reported in the second line treatment of *BRAF^V600E^* mutant mCRC patients. Ramucirumab-a highly specific antiangiogenic agent directed against the extracellular domain of the *VEGF* receptor-2-may block the activating phosphorylation of the proangiogenic receptor. In the VELOUR trial, a subgroup analysis showed that 11 patients treated with FOLFIRI plus aflibercept had a median PFS and OS compared with 19 patients receiving only chemotherapy for 5.5 and 10.3 months vs. 2.2 and 5.5 months, respectively [61]. Similar results were also observed with FOLFIRI plus ramucirumab in the subgroup analysis of the RAISE study [62]. However, the value of these post-hoc analyses should be carefully considered given the small number of patients included. 

Unlike the favorable results observed in melanoma patients, treatment with *BRAF* inhibitors alone has yielded low clinical activity in mCRC due to feedback reactivation of *EGFR* [63]. Therefore, *BRAF^600E^* inhibitors have been combined with anti-*EGFR* inhibitors and other targeted agents, such as *MEK* inhibitors, in doublet and triplet regimens with the aim of overcoming resistance to *BRAF* inhibitor monotherapy and improving its clinical activity [64,65,66,67,68].

This strategy was evaluated in the phase III BEACON study in which 665 patients with *BRAF^V600E^* mutated mCRC and disease progression after one or two prior treatments were randomized to receive the *BRAF* inhibitor encorafenib in combination with the anti-*EGFR* antibody cetuximab plus the *MEK* inhibitor binimetinib (triplet therapy) versus encorafenib and cetuximab (doublet therapy) or cetuximab in combination with irinotecan or FOLFIRI as standard therapy [69]. The primary endpoints of the study were OS and RR. In the primary analysis, the triplet and doublet therapy compared with standard therapy resulted in a significantly longer median OS (9.0 and 8.4 months vs. 5.4 months, respectively, *p* < 0.001) and higher RR (26% and 20% vs. 2%, respectively, *p* < 0.001). The PFS, secondary endpoint, was also superior in the three-drugs and doublet-drugs regimen when compared to the standard group (4.3 and 4.2 months vs. 1.5 months, respectively, *p* < 0.001). In addition, safety and tolerability were not significantly different between the groups, resulting in grade 3 or higher AEs in 58% of cases in the triplet regimen, in 50% of cases in the doublet regimen, and in 61% of patients treated with standard therapy, respectively. The updated results of this study, with additional six months of follow up, showed that an OS of 9.3 months was observed for both triplet and doublet regimens compared to 5.9 months of the control therapy [18]. These results strongly support the combination of encorafenib plus cetuximab as a new standard of care for pretreated mCRC *BRAF^V600E^* mutated patients. A single arm phase II trial (ANCHOR study), evaluating the strategy with triplet regimen in the first-line setting, which has recently been completed, reported findings similar to the BEACON study [70]. Finally, on the basis of these results, the ongoing phase III study BREAKWATER (NCT04607421) aims to evaluate the efficacy in terms of PFS (primary endpoint) and OS of the combination of encorafenib plus cetuximab with or without chemotherapy compared with a standard anti-VEGF based treatment in the 1st line setting.

## 5. Microsatellite Instability and Immunotherapy

Immunotherapy in cancer treatment arises from the concept that a condition of immunoevasion exists caused by neoplastic cells in the tumor microenvironment. Tumor cells, through the production of cytokines, stimulate suppressor myeloid cells and regulatory T cells (*Treg*) to inhibit the CD4+ and to increase CD8+ lymphocytes, braking immune responses. Furthermore, a loss of restricted major histocompatibility complex molecules has also been observed, resulting in an inability of the host to recognize non-self-antigens [71].

Microsatellites are repetitive sequences of coding, and non-coding *DNA* [72]. *MSI* results from the inability of the *MMR* gene to repeat *DNA* errors that occurred during the replication process. Gene insertions and deletions lead to somatic mutations in these repetitive *DNA* sequences resulting in genomic instability and production of immunogenic antigens and neoantigens, conditioning a response to checkpoint inhibitors [23]. Inactivation of *MMR* genes is the result of hypermethylation of the *MLH1* promoter or of germline mutations of *MLH1*, *MSH2*, *MSH6,* and *PMS2* [73]. 

Furthermore, *MSI* germline abnormalities also represent the molecular basis of Lynch syndrome [74]. It represents the most common hereditary form of this cancer. Latham et al. reported that *dMMR* is common in these patients, so those with *MSI* or *dMMR* tumors could predict Lynch syndrome through *MSI* related tests [75]. 

*MSI* is found in approximately 5% of patients with mCRC; only 3% of cases are associated with Lynch syndrome and the other 12% are caused by sporadic hypermethylation of the *MLH1* gene. CRC with *MSI* are most frequently localized on the right and in women over 70 years, are poorly differentiated, and have mucinous histology [76]. *MSI* tumor status could be a prognostic marker for a more favorable outcome. A large study reported that the percentage of mCRC patients with this characteristic was only 3.5% suggesting that these tumors have a lower probability of metastasizing [77]. 

The incidence of *MSI* in stage II and III is about 16%. Some studies have shown that *dMMR* or *MSI* tumor status are predictors of reduced benefit from adjuvant chemotherapy and that fluoropyrimidines given alone may even have a detrimental effect in patients with stage II disease [78,79,80]. Conversely, regarding patients with *MSI* and stage III, ACCENT, a pooled analysis of 12 adjuvant studies, has demonstrated that adding oxaliplatin to fluoropyrimidines improves DFS and OS of patients compared to those of stable microsatellite tumors (*MSS*) [81]. In particular, this study found a close relationship between the number of positive lymph nodes on histological examination and OS, documenting better outcomes in the N1 group, while data were similar in the N2 group.

In the metastatic setting, a phase II study demonstrated that tumors with a high mutation load benefit most from the use of pembrolizumab. The primary endpoints were immune-related ORR and PFS at 20 weeks. Forty-one patients with progressive disease had ORR and PFS at 20 weeks of 40% and 78% for *MSI* tumors and 0% and 11% for MSS ones, respectively [82].

CheckMate-142, another phase II study, reported efficacy and safety results in 119 patients with *MSI* status and previously treated. They received nivolumab 3 mg/kg plus ipilimumab 1 mg/kg once every 3 weeks (four doses) followed by nivolumab 3 mg/kg once every 2 weeks. Primary end point was investigator-assessed ORR. At the median follow-up of 13.4 months, investigator-assessed ORR was 55% (95% CI, 45.2 to 63.8), and DCR for ≥ 12 weeks was 80%. Regarding the median duration of response, more than half of the patients were still responding, so the median is unknown, while PFS rates were 76% (9 months) and 71% (12 months); respective OS rates were 87% and 85%. Grade 3 or 4 AEs were manageable and occurred in 32% of patients [19]. Nivolumab plus ipilimumab could represent a promising new treatment option in these patients.

Finally, in the phase III Keynote 177 trial, pembrolizumab resulted in significantly longer PFS than chemotherapy when received as first-line therapy for *MSI* mCRC [13]. Co-primary endpoints were PFS and OS. Three hundred and seven patients were assigned to receive pembrolizumab at a dose of 200 mg every 3 weeks or investigator’s choice standard chemotherapy every 2 weeks. Patients undergoing chemotherapy could receive pembrolizumab after disease progression. After a median follow-up of 32.4 months, pembrolizumab was superior to chemotherapy with respect to PFS (median, 16.5 vs. 8.2 months; HR, 0.60; *p* = 0.0002). At the cutoff date, 56 patients in the pembrolizumab group and 69 in the chemotherapy group had died. OS data were not yet mature after final data evaluation and are pending. An ORR was observed in 43.8% of the patients in the pembrolizumab group and in 33.1% in the chemotherapy group. In addition, the responses appeared to be more prolonged over time with immunotherapy. The AEs of grade 3 or higher occurred in 22% of the patients in the pembrolizumab group, as compared with 66% in the chemotherapy group. A subsequent analysis evaluating the impact of therapy on quality of life showed clinically meaningful improvement in EORTC QLQ-C30 GHS /QOL scores with pembrolizumab compared to chemotherapy [83]. 

## 6. *HER2* Inhibition

The *HER* family plays a crucial role in the development and progression of several gastrointestinal tumors, including colorectal, gastric, and biliary adenocarcinomas; its aberrant activation-mainly due to overexpression via *HER-2* gene amplification or to alternative genetic mechanisms-has been reported consistently in 5–20% of cancer patients [84,85]. The possibility of inhibiting *HER-2* to tackle the progression of the disease is certainly not new, and pivotal randomized trials have shown that the use of trastuzumab either alone or combined with another *HER-2* blockade agent has significantly extended survival in molecularly selected cases [86,87,88]. In addition, it stimulated the need for specific classifications and scoring systems to establish *HER-2* positivity [20], which is usually scored with immunohistochemistry (*IHC*), and then confirmed with in situ hybridization or innovative, more sensitive techniques [89]. The *IHC* scoring system for *HER-2* positivity in CRC was established by experienced pathologists involved in the HERACLES project. In more detail, *IHC* staining judged as intense (3+) in more than 10% of cancer cells with circumferential, basolateral, or lateral pattern was defined as positive; the expert panel recommended to confirm the positivity if the percentage of positive cells was inferior to 50% [20]. As an outstanding example for gastrointestinal oncology, in the open-label, multicenter, international, phase III ToGA trial the combination of standard chemotherapy and trastuzumab was compared to chemotherapy alone [86]. In patients with *IHC* 3+ *HER-2*-positive advanced gastric cancers treated with trastuzumab the reported median OS was about 4 months longer that that reported for patients treated with standard therapy (16.0 versus 11.8 months, HR 0.65), and the drug gained accelerated Food and Drug administrative approval. In mCRC, *HER-2* has been shown to represent a notable therapeutic target, regardless of its primary or acquired resistance to *EGFR* inhibition [90,91], although prognostic impact of *HER-2* overexpression/amplification has not yet been fully elucitated [92]. HERACLES, a proof-of-concept phase II academic trial, enrolled 35 *HER-2*-positive, *RAS* wt, mCRC patients refractory to standard therapies (including cetuximab or panitumumab), with 32 patients evaluable for response. Enrolled patients received intravenous trastuzumab at 4 mg/kg loading dose followed by 2 mg/kg once per week, and oral lapatinib at 1000 mg per day until evidence of disease progression. A RR of 28% was reported with one case of complete response lasting over 7 years, a DCR of 69%, a median PFS of 4.7 months (95% CI 3.7–6.1), and a median OS of 10 months (95% CI 7.9–15.8) [14,89]. Interestingly, progression in the central nervous system occurred in 6 (19%) out of 32 patients, suggesting that the evaluation of *HER-2* expression in brain metastases from CRC is important [93]. More recently, the improved understanding of *HER-2*-driven disease pathology in specific gastrointestinal cancers [94], the comprehension of complex primary or acquired resistance mechanisms [95,96,97,98], the development of novel, more potent *HER-2* inhibitors [99], and the possibility to synergistically combine *HER-2* inhibition with immunomodulating treatment strategies [100,101], have all contributed to a significant evolution of the treatment scope. In fact, the ultimate frontier for modern *HER-2* inhibition in CRC includes three distinctive therapeutic roads. 

Firstly, the use of specific antibody-drug conjugates, one of the foremost being trastuzumab deruxtecan (T-DXd), a humanized anti-*HER-2 IgG1* monoclonal antibody covalently linked to a topoisomerase I inhibitor payload via a tetrapeptide-based cleavable linker. In the Destiny-CRC01 trial [21], this novel drug demonstrated promising and durable activity in patients with *HER-2*-positive *mCRC* refractory to available standard treatments, with an ORR of 45% and a median PFS of almost 7 months. Since antibody-related lung toxicity (i.e., interstitial lung disease/pneumonitis) may affect up to 10% of treated patients [102], this worrying side-effect should be considered and promptly assessed at least in symptomatic patients [100]. Secondly, the study of new specific monoclonal antibodies is providing novel treatment opportunities. Margetuximab, a next-generation Fc-modified *HER-2* monoclonal antibody that binds with high affinity to CD16A, has activity in *HER-2* positive gastrointestinal malignancies [103]. Lastly, while evidence from studies combining two different antibodies in pretreated patients is rapidly accumulating [104,105,106,107], bi-specific antibodies that may simultaneously target two different parts of *HER-2* could potentiate their effect. In HERACLES-B, a single-arm, phase II trial, enrolling highly pretreated patients with *RAS*/*BRAF* wt and *HER-2* positive mCRC, the combination of pertuzumab (840 mg intravenous load followed by 420 mg intravenous every 3 weeks) and T-DM1 (3.6 mg/kg every 3 weeks) was evaluated until disease progression or toxicity [104]. Unfortunately, the trial did not reach its primary endpoint, with a disappointing RR of 9.1% and a median PFS of 4.8 months. Three studies–namely My Pathway, TRIUMPH, and TAPUR-tested trastuzumab in combination with pertuzumab, with a RR ranging from 25% to 40% and median time to progression (MTP) of 4 to 5 months. In MOUNTAINEER, the combination of trastuzumab and tucatinib produced a RR of 52% and a MTP of 8.1 months, suggesting a synergistic value for the combination. In the meantime, zanidatamab (ZW25), a novel biparatropic targeting drug that can bind to two different *HER-2* epitopes, showed high antitumoral activity in a phase I trial gastrointestinal adenocarcinomas with *HER-2* overexpression [108]. Confirmatory trials have already started. In this evolving scenario towards the concept of precision oncology, there is the need of appropriate patient selection. A more profound molecular characterization established upfront [109] or at the time of disease progression [110], a better understanding of the optimal treatment sequences [111], the possibility to drive targeted treatments with easily repeatable tests [112], and the development of novel, potent inhibitors [113], will shape the near future and lead forthcoming clinical improvements in *HER-2* positive CRC tumors.

## 7. Targeting *NTRK*, *ALK,* and *ROS1* Fusions

Among novel actionable targets in mCRC, gene fusions such as *NTRK* rearrangements or fusions of anaplastic lymphoma kinase (*ALK*) or Proto-Oncogene 1 Receptor Tyrosine Kinase (*ROS1*) are of growing importance [114]. While several pathogenic alterations have been reported for such genes, including point mutations, amplifications, and splice variants, fusions are the most common genetic aberrations linked to cancer and cause constitutive gene activations and hyper-activation of the kinase domain. In mCRC, these fusions/rearrangements are rare (0.5–2%) and most frequently occur in elderly patients with right-sided, lymph-node positive, *RAS* wt, *MSI* cancers. They may suggest resistance to *EGFR*-inhibitors, have a negative prognostic survival impact and may be targeted with specific agents [115]. Larotrectinib and entrectinib are oral tropomyosin receptor kinases (*TRK*). Upon administration, these agents bind to *TRK*, preventing neurotrophin-*TRK* interaction and *TRK* activation, which results in both cellular apoptosis and the inhibition of cell growth in tumors that overexpress *TRK*. Based on the impressive results of agnostically testing larotrectinib and entrectinib in cancer patients with *NTRK* rearrangements, with a very high ORR in molecularly selected cases [116,117,118] and improvements in cancer-specific quality of life [115], both agents gained the Food and Drug Administration and the European Medicine Agency approval. In the phase II NAVIGATE study, larotrectinib produced an ORR of 50%, with a median duration of response of 15.5 months, and median OS of almost 30 months [22]. The possibility to use third-generation *ALK* inhibitors in mCRC has been suggested [119], but the rarity of this gene alteration makes it difficult to conduct large comparative trials. Novel *NTRK*/*ROS1* inhibitors, including selitrectinib, repotrectinib, and belizatinib, are under investigation in early clinical trials [120].

## 8. Conclusions

As with other cancers, genetic heterogeneity and the consequent stratification of patients is now a reality in the treatment of mCRC. Using in-depth knowledge of the molecular asset, each patient therefore has the opportunity to receive a therapy, even more targeted than was possible until a few years ago through the simple determination of the status of *RAS* and *BRAF*. To treat mCRC properly, personalized oncology will soon include many tests useful for the identification of novel putative targets to further enrich the growing toolbox of agents for mCRC patients. It is to be hoped that the progress of molecular biology will be even more tangible in the coming years, even if this presupposes the establishment of adequately equipped laboratories in order to optimize the economic resources of each national health service. 

## Figures and Tables

**Figure 1 biomedicines-10-01035-f001:**
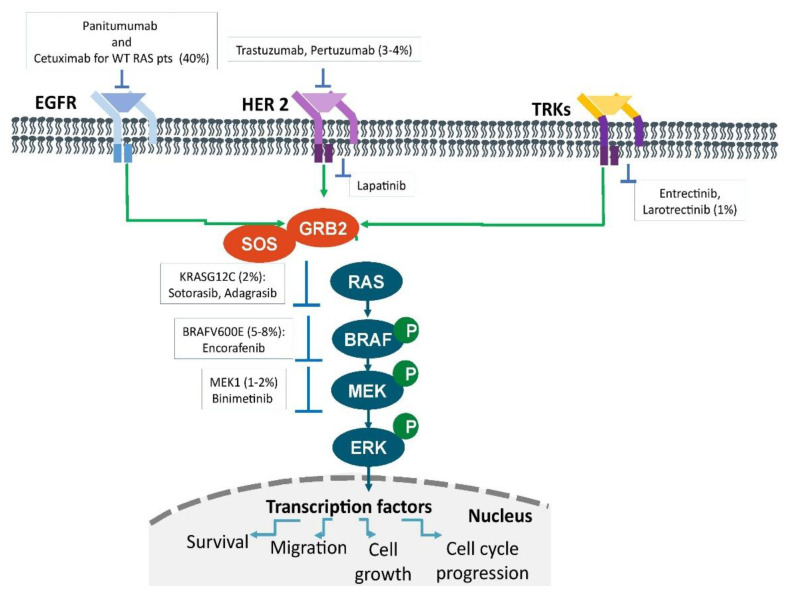
Therapeutic targets in metastatic colorectal cancer. The main oncogenic drivers. Signaling pathways and their prevalence in patients with metastatic colorectal cancer.

**Table 1 biomedicines-10-01035-t001:** Selected trial data for emerging biomarkers in mCRC.

Biomarker	References	No. of Patients	Treatment	Key Findings
*KRAS^G12C^*	[16]	42	Sotorasib	Median PFS 4.0 months; PR 7.1%; SD 66.7%
	KRYSTAL-1 [17]	28	Adagrasib + cetuximab	ORR 43%; DCR 100%
*BRAF*	BEACON [18]	665	Encorafenib + Cmab vs. Encorafenib + Cmab + binimetinib vs. Cmab + CT	Median OS (9.0 and 8.4 months vs. 5.4 months, *p* < 0.001); ORR (26% and 20% vs. 2%, *p* < 0.001)
*MSI*	CheckMate-142 [19]	119	Nivolumab + ipilimumab	ORR 55%; DCR > 12-week of 80%
	Keynote 177 [13]	307	Pembrolizumab vs. CT	PFS 16.5 vs. 8.2 months, *p* = 0.0002; ORR 43.8% vs. 33.1%
*HER-2*	HERACLES [14,20]	32	Trastuzumab + lapatinib	ORR 28%; DCR 69%; median PFS 4.7 months
	Destiny-CRC01 [21]	53	Trastuzumab deruxtecan	ORR 45%; median PFS 7 months
*NTRK*	NAVIGATE [22]	40	Larotrectinib	ORR 50%; median OS 30 months

*Cmab* cetuximab, *CT* chemotherapy, *DCR* disease control rate, *mCRC* metastatic colorectal cancer, *ORR* overall response rate, *OS* overall survival, *PFS* progression-free survival, *PR* partial response, *SD* stable disease.

## Data Availability

Not applicable.

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
