# Peer review of "Colorectal Cancer Heterogeneity and the Impact on Precision Medicine and Therapy Efficacy"

_biomedicines, 2022, doi:10.3390/biomedicines10051035_

Round 1

Author Response

Attached are the comments and changes made to the manuscript.

Reviewer 2 Report

Very well-written and informative review paper. In the introduction section, I would recommend a paragraph describing the authors' search method to find the cited literature (e.g., keywords, databases, search period, etc.). Also sentence "The review focuses on three themes: evolution of molecular biology in mCRC patients, development of mechanisms of resistance to chemotherapy and anti-EGFR and anti-VEGF monoclonal antibodies and differential effect of new targeted therapies." is not 100% accurate, and I would recommend changing with a more vague statement. Finally, I could not see figure 1, although it is mentioned in the manuscript.

Author Response

Attached are the comments and changes made to manuscript.

Reviewer 3 Report

Comments to the Review: Colorectal Cancer Heterogeneity and the Impact on Precision 2 Medicine and Therapy Efficacy

The manuscript presented by Rosati et al. is a broad description of current treatments undergoing in the clinics to treat metastatic colorectal cancer. The review is up to date with the last clinical trials and pleasantly summarize the results from each of them. It can be recommended to new Scientifics in the field of colorectal cancer to understand the complexity of this type of neoplasia.

It is a well-structured and easy to follow, however there are major a minor comments to address:

Major comments

Immune system has been poorly described along the review and is a major player in therapeutic responses and resistances. I will highly encourage the authors to take into consideration, especially in chapter 4 with MS description.

Line 26: Abstract conclusion is imprecise. I will considerer to make a more accurate statement.

Line 34: epidemiology should described actual time frame, 2018 is four years ago and WHO and ECIS (European Cancer Information System)

The utility of the review will be high-graded if the authors summarize key points from each chapter in a table, addressing Clinical trial name, mutation of study, sample size, treatment arms, PFS/OS and reference. This can be addressed in the conclusion part, which does not summarize the molecular complexity of CRC whereas described a wishing feeling from the authors.

Minor comments:

  • Figure 1 is confusing, neutralizing antibodies and inhibitors are considered equally, I will encourage the author to address the drugs in the figure.
  • Line 189-192 need the reference for such statements.
  • Although FOLFOX, FOLFIRI and FOLFOXIRI are well known frontline strategies, it needs to be characterized at least once, mentioning the chemotherapy agents involved in the protocol.
  • Missing molecular target for Ramucirumab, Larotrectinib, Entrectinib

Author Response

(The authors gave the same response as above.)
